# High-performance energy harvesting and continuous output using nylon-11/BaTiO₃-PVDF triboelectric nanogenerators with strong dielectric properties

Xiong Dien[1,2,3], Nurulazlina Ramli[2], Tzer Hwai Gilbert Thio[2], Cheng Linfeng[3], Zhang Lin[4], Yuan Jiang[3], Yang Zhuanqing[1]*

1 School of Big Data and Internet of Things, Chongqing Vocational Institute of Engineering, Chongqing, China, 2 Faculty of Engineering, Built Environment and Information Technology, SEGi University, Kuala Lumpur, Malaysia, 3 Chongqing Vocational and Technical University of Mechatronics, Chongqing, China, 4 School of Electronic Information Engineering, Geely University of China, Chengdu, China

* yangzq@cqvie.edu.cn

## Abstract

Among various emerging energy technologies, triboelectric nanogenerators (TENGs) have garnered significant attention owing to their ability to convert environmental mechanical energy into electrical energy through the triboelectric effect and electrostatic induction. However, there are some problems with optimizing the electrical output and conversion efficiency of TENGs. This paper presents a high-performance TENG enhanced with BaTiO₃ nanowires(BTONWs) using electrospinning technology. PVDF was doped with BTONWs to fabricate TENGs with high flexibility and efficient energy conversion. BaTiO₃ and PVDF all exhibited inherent properties and triboelectric properties, maximizing the conversion of pressure into electrical energy output. This integration effectively enhances the conversion power and provides a continuous energy supply. Experimental results show that the fabricated TENGs achieved a current and voltage of 12 μA and 280 V, respectively, with a maximum power density of 1.45 W/m² at a load resistance of 90 MΩ. In addition, the performance of the TENGs was tested using a calculator, timer, and LED lights. By connecting to a simple external circuit and continuously tapping the TENG, the devices functioned normally, demonstrating that the TENG can constantly and stably output electrical energy by continuously collecting mechanical energy to power microgenerators. This work may significantly contribute to developing energy harvesting, wearable devices, and micropower sources.

## Introduction

The rapid development of society has highlighted the critical importance of clean and renewable energy. Renewable energy sources, such as triboelectric

**Data availability statement:** All relevant data are within the manuscript.

**Funding:** This work was supported by Natural Science Foundation Project of Chongqing Science & Technology Commission through grant No.2024NSCQ-MSX4013, and Scientific and Technological Research Program of Chongqing Municipal Education Commission through grant No.KJQN202403414.

**Competing interests:** The authors have declared that no competing interests exist.

nanogenerators (TENG) and solar, wind, hydro, and bioenergy, offer hope for a sustainable future while reducing environmental impacts [1]. However, owing to the intermittent and variable nature of these renewable energy sources, there is a need to develop new technologies for their efficient utilization. Among various emerging energy technologies, TENGs have attracted attention for their ability to convert ambient mechanical energy into electrical energy through the triboelectric effect and electrostatic induction [2]. TENGs can harvest mechanical energy from various sources, including human movement, vibration, wind, and water waves. The triboelectric effect is a phenomenon in which certain materials become electrically charged upon frictional contact with another material. This effect, combined with electrostatic induction, enables TENGs to generate electrical energy via mechanical motion [3]. This makes TENG a versatile and promising technology for energy-harvesting applications. However, TENGs limit their use owing to their lower output power density, and their low conversion efficiency makes it difficult to drive loads [4].

In recent years, with the development of TENG technology, researchers have attempted to improve the performance of TENGs by improving composite materials, surface structures, and fabrication methods [5]. The selection of materials is critical to the performance of TENGs because the triboelectric effect depends on the properties of the materials. For example, the addition of zinc oxide (ZnO) to polyvinylidene difluoride (PVDF) improves the electrical properties of the material, thereby increasing its electrical output [6]. Meanwhile, as a material to change the electrical properties, $BaTiO_3$ improved the dielectric constant of the composites more significantly than ZnO. $BaTiO_3$ promoted better crystallization of PVDF. In addition to its good electrical properties, $BaTiO_3$ has better performance as a composite [3]. However, polydimethylsiloxane (PDMS) has strong adhesive properties and cannot be easily electrospinning [7]. Most of the TENGs made from PDMS were made by molding, resulting in larger diameter fiber filaments [8] or wrapped with other substances before electrospinning [9]. In contrast, PVDF can be directly electrospun electrostatically, which is simpler to fabricate and has a smaller imaging diameter.

Specifically, this paper presented the fabrication of a new TENG of high-performance, highly flexible using nylon-11 and PVDF with $BaTiO_3NWs$. $BaTiO_3NWs$ was mixed with PVDF and then, electrospun into a triboelectric layer. PVDF and $BaTiO_3NWs$ were chosen because of their excellent triboelectric properties, which can synergistically enhance the energy output of TENG. $BaTiO_3NWs$ with different mass ratios were investigated and analyzed to determine the optimal composition to achieve maximum electrical output. The electrospinning process was optimized to produce a homogeneous and highly porous structure, which is essential for increasing the contact area and improving the TENG efficiency.

This study aimed to address the existing technical and application problems through the development of novel, efficient, and eco-friendly TENG systems. By exploring innovative materials and design approaches, the objective was to improve the energy conversion efficiency and environmental resilience of TENG and to

evaluate its performance stability in practical applications. The final goal was to pave the way for the commercialization and sustainable development of TENG technology in energy harvesting.

Specifically, this paper presented the fabrication of a new TENG of high-performance, highly flexible using nylon-11 and PVDF with BaTiO₃NWs. BaTiO₃NWs was mixed with PVDF and then, electrospun into a triboelectric layer. PVDF and BaTiO₃NWs were chosen because of their excellent triboelectric properties, which can synergistically enhance the energy output of the TENG. BaTiO₃NWs with different mass ratios were investigated and analyzed to determine the optimal composition to achieve maximum electrical output. The electrospinning process was optimized to produce a homogeneous and highly porous structure, which is essential for increasing the contact area and improving the efficiency of TENG.

In the experiment, we fabricated and tested an electrospun TENG of BTONW-PVDF. The results showed that, at a load resistance of 90MΩ, the TENG achieved a current of 12µA and a voltage of 280V, with a maximum power density of 1.45 W/m$^2$. This indicates that the TENG has a high energy-conversion efficiency and can generate a significant amount of electrical energy from mechanical motion. In addition, to verify the practical application power of the TENG, we designed a series of experiments using a calculator, timer, and LED lights as testing objects. By connecting the TENG to an external circuit and continuously patting the TENG, the calculator and timer operated normally and the LED lights were illuminated. This demonstrates that the electrical energy generated by the TENG made of BTONW–PVDF can be fully applied to power microelectronic devices. This study aims to significantly advance the development of TENG technology, contributing to the realization of sustainable energy solutions and reduction of environmental impacts. Addressing technical challenges and exploring new wearable applications.

## Experiment

### Materials

Nylon-11 and poly (vinylidene difluoride) (PVDF, FR 904) (Mw = 600000) were purchased from Aladdin Shanghai 3F New Material Co., Ltd. Dimethylformamide (DMF), acetone (AR), Titanium Dioxide, Barium hydroxide octahydrate was obtained from the Chengdu Kelong Chemical Reagent Factory. The 1,1,1,3,3,3- Hexafluoro- 2- propanol (HFIP) was supplied by Aladdin. Nylon-11 pellets were obtained from Sigma-Aldrich. All agents were used directly.

### Preparation of BTONWs

Hydrothermal reaction was used to prepare the BTONWs [10]. First, Na2Ti3O7 NWs were prepared as an intermediate product. 2 g of titanium dioxide was added into 40 ml of 10 M NaOH aqueous solution with stirring for 1 h. The solution was then poured into a 250 ml Teflon autoclave and heated at 200 °C for 72 h. The intermediates were collected, washed with deionized water and alcohol several times, and dried in a vacuum oven at 60 °C. Second, the intermediates (0.5 g) were added to 10 mL of deionized water with stirring for 1 h. Subsequently, the mixture was poured into 40 ml of 0.12 M barium hydroxide solvent and then strongly agitated for 1 h. The solution was poured into a Teflon autoclave and maintained at 100 °C for 24 h. After drying in a vacuum oven at 60 °C, the BTONWs were obtained.

### Electrospinning and fabrication of TENG

As illustrated in Fig 1 (a)–(d), the as-prepared BTONWs (0 wt%, 1 wt%, 2 wt%, and 3 wt%) were added to the mixture solution (DMF/ACE = 6/4, w/w) and ultrasonicated for 2 h to disrupt any entangled BTONWs. PVDF powder was added to the solvent and stirred for 2 h in an oil bath at 60 °C to obtain a precursor concentration of 10 wt%. Similarly, 1.11 g Nylon-11 pellets were added to 10 g of HFIP and stirred for 2 h. A homemade electrospinning setup was used to fabricate nanofibers. The precursors into a 10 ml syringe and pumped at a feeding rate of 1 mL/h with a 20 G needle. Electrospinning was performed at 21 KV, with collecting distances of 17 cm, and the rotating drum was set at 1500 rpm. Finally, BTONW-PVDF, pristine PVDF, and Nylon-11 nanofibers were obtained on aluminum foil.

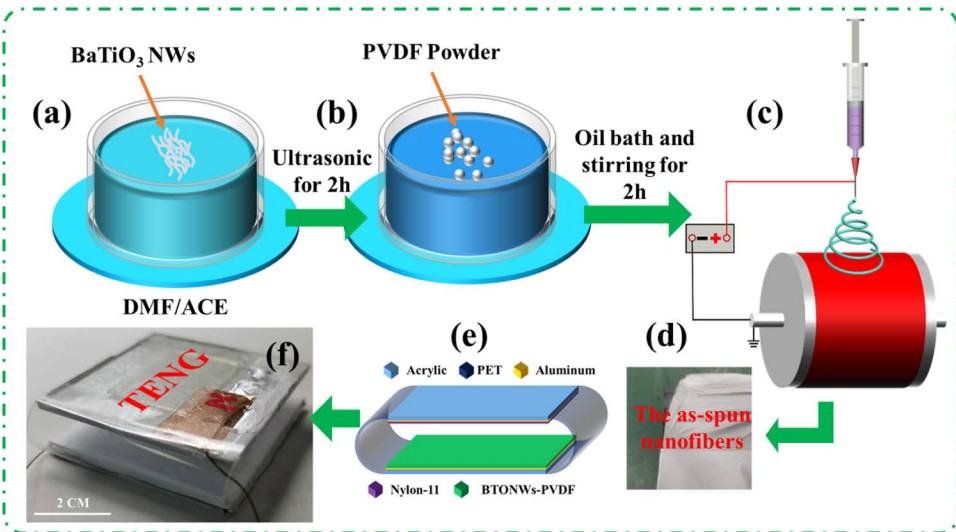

**Fig 1. (a)–(d) Preparation of the BTONW-PVDF nanofibers.** (e) Design of TENG. (f) Digital picture of the assembled TENG.

To assemble the TENG, BTONW-PVDF nanofibers were first deposited on an aluminum foil using electrostatic spinning, and the resulting sample was cut into 4x4 cm² squares. Subsequently, the squares were fixed to laser-cut 3 mm thick acrylic substrates using double-sided adhesive tape to tightly fit the side with the nanofibers (the aluminum foil as the backing layer) to the acrylic plate, so that the aluminum foil served as both a mechanical support layer and an electrode. To induce an electrical signal, a copper wire with a diameter of about 0.51 mm was connected to aluminum foil, and a transparent PET film with a thickness of 0.5 mm was used as a contact-separation friction interface (as shown in Fig 1). Finally, the copper wire was connected to a Keithley Model 6514 electrical test system to characterize the electrical output of the device, including current and voltage. This designed structure not only simplifies the device assembly steps but also lays the foundation for subsequent performance evaluation and application testing.

## Characterization

Scanning electron microscopy (SEM, FEI Inspect F50) was used to investigate the surface morphology, cross-sectional morphology, and elemental distribution of the prepared friction layer and mixture. Transmission electron microscopy (TEM, FEI G2 F20) was used to investigate the crystallinity and doping characteristics of the prepared friction layers and mixtures. X-ray diffraction (XRD, X' Pert Pro MPD DY129) was used to characterize the friction layer and mixture. Fourier-transform infrared spectroscopy (FT-IR, NEXUS-670) was used to characterize the infrared absorption of nylon-11 and its mixture. X-ray photoelectron spectroscopy (XPS, AXIS Supra). The electric voltage and current outputs were acquired using a Keithley 6514.

## Results and discussion

Fig 2 shows a schematic of the device and the fabricated TENG. When nylon-11 and PVDF-BTONWs are in contact under pressure, a surface charge transfer occurs in the contact area owing to the triboelectric effect, where the positively charged layer of nylon-11 (top) loses electrons to carry a positive charge, while the negatively charged layer of the PVDF-BTONWs (bottom) gains electrons to carry a negative charge. After the separation of nylon-11 and PVDF-BTONWs, a potential difference between the top and bottom electrodes occurs because of electrostatic induction, which induces electrons to flow from the bottom electrode to the top electrode, forming a half-cycle negative current pulse. When nylon-11

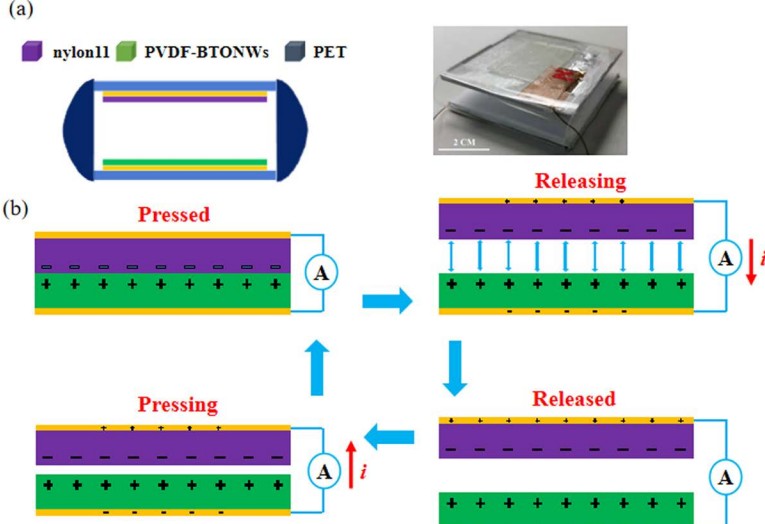

**Fig 2. Schematic of the structure and working mechanism of TENG.**

and PVDF-BTONWs are pressed, a reverse current pulse, that is, a half-cycle positive current pulse, is generated. Repeatedly, the external circuit can generate AC during each pressing and releasing cycle.

A physical image of the TENG fabricated using nylon11/BTONWs -PVDF is shown in Fig 3 (a). Fig 3(b) shows the SEM image of the nylon 11 fibers. The nylon 11 fibers exhibited a homogeneous network structure with relatively uniform fiber diameters, which were tightly aligned and had no obvious breaks or defects. This homogeneous fiber network helped to increase the surface area of the material, thus enhancing the friction electric effect. The smoothness and continuity of the fiber surface indicated that better process parameters were controlled during the preparation process to ensure the integrity and consistency of the fibers. Fig 3(c) shows the SEM image of the negative friction electric layer BTONWs-PVDF. As shown in the image, the BTONW-PVDF fibers formed a denser network structure, with the fibers interleaved with each other, constituting a complex three-dimensional network. Compared with the nylon 11 fibers, the BTONW-PVDF fibers had a slightly different diameter distribution and a slightly rougher surface. This rough surface property contributed to the friction electrical properties of the fibers, as the larger surface area increased charge generation and accumulation. The interwoven structure of the fibers ensured the mechanical stability and durability of the material while enhancing its potential as a friction material for TENG. These structural features enabled the BTONW-PVDF fibers to exhibit excellent performance in energy harvesting and conversion. Fig 3(d) shows SEM images of the PVDF fibers doped with different mass ratios of BTONWs. The PVDF fibers doped with 1 wt% BTONWs formed a uniform and dense network structure, and the fibers were interspersed with each other to form a stable three-dimensional structure. The PVDF fibers doped with 2 wt% BTONWs had a denser network and increased surface area compared to those doped with 1 wt% BTONWs, indicating that more BTONWs were successfully doped into the fibers. The structure of the PVDF fibers doped with 3 wt% BTONWs was denser than the first two doping ratios, indicating that more BTONWs were contained within the fibers. Fig 3(e) shows a scaled SEM image of different doping ratios of BTONWs up to 1 μm. The PVDF fibers doped with 1 wt% BTONWs show the details of the fibers. The surface of the fibers was slightly rough, and the BTONWs were uniformly distributed in the PVDF fibers, which enhanced the mechanical strength and electrical properties of the material. The PVDF fibers doped with 2 wt% BTONWs showed a higher percentage of BTONWs uniformly distributed inside the fibers, with a rougher fiber surface. This roughness further enhanced the friction electrical properties and helped to improve the electrical output of TENG. The PVDF fibers doped with 3 wt% BTONWs demonstrated a highly doped fiber structure. Although

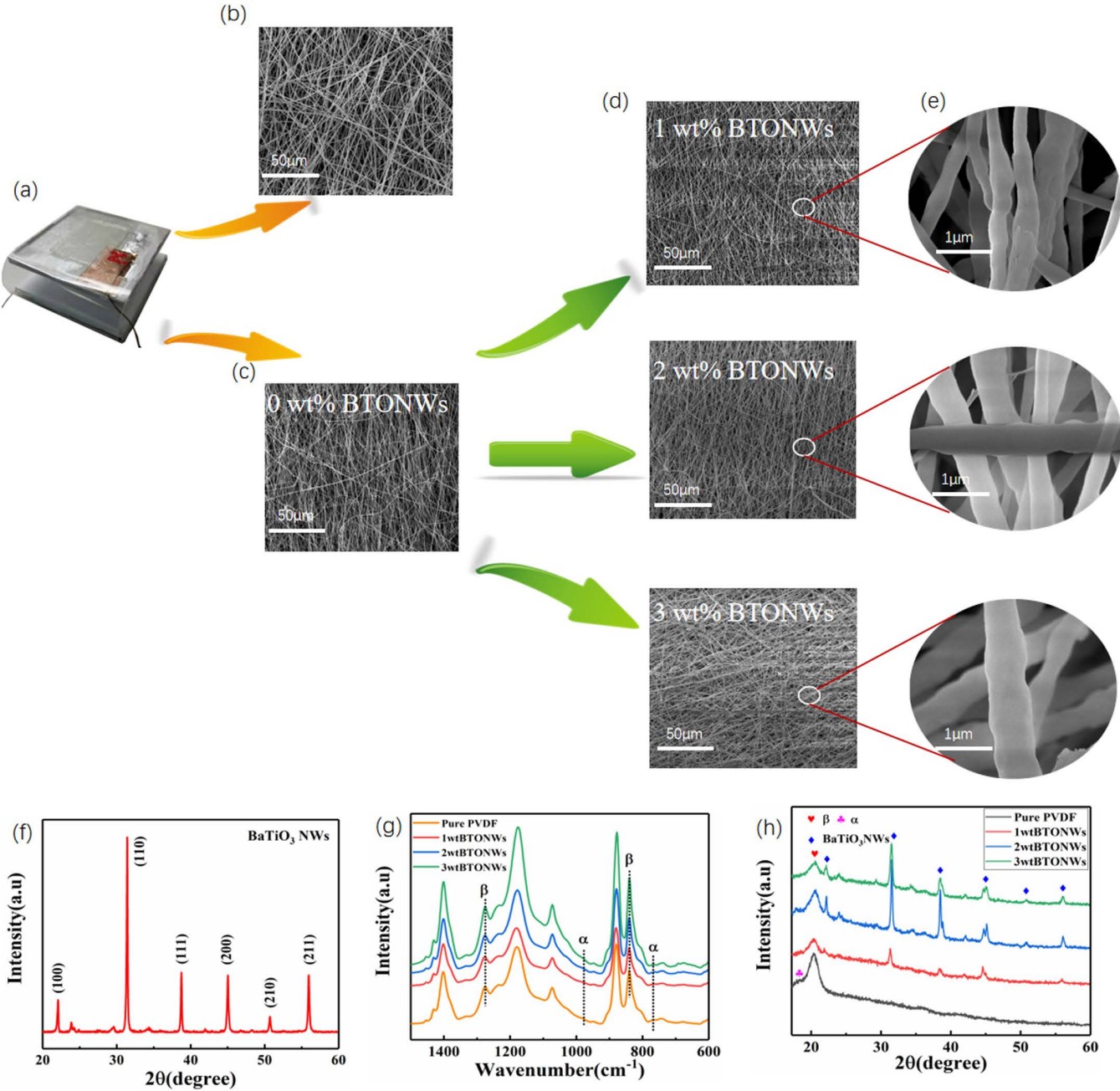

**Fig 3. Analytical image of TENG produced by nylon 11 and PVDF-BTONWs. (a)** Physical image of TENG. **(b)** Surface morphology of nylon-11 fibers. **(c)** Surface morphology of BTONWs -PVDF. **(d)** SEM images of BTONWs doped with different mass ratios. **(e)** Magnified images of BTONWs doped with different mass ratios. **(f)** XRD image of BTONWs. **(g)** XRD patterns of PVDF, BTONWs-PVDF film, and BTONWs before mixing with PVDF. **(h)** FTIR absorption spectra of BTONWs-PVDF nanofibres with different BTONWs mass fractions.

the surface roughness and the distribution of BTONWs increased further, excessive doping might lead to a decrease in the mechanical properties of the fibers. The surface roughness and structural density of the PVDF fibers increased as the percentage of BTONWs doping increased. These changes contributed to the improvement of the friction electrical properties, but there was also a need to find a balance between the mechanical and electrical properties to optimize the overall

performance of the TENG. The XRD pattern of BTONWs is illustrated in Fig 3(f), which shows the main diffraction peaks of BTONWs. Several distinct diffraction peaks were labeled in the Fig, which appear in the range of 20° to 60°, corresponding to different crystal planes of the barium titanate crystals. The strongest diffraction peak was obtained from the (110) crystal plane, which appears at about 31.5 degrees, showing that BTONWs have the strongest crystal orientation at the (110) crystal plane, and indicating the high crystallinity and tetragonal phase of BTONWs [11]. The clarity and intensity of the peaks showed that the BTONWs have good crystal quality and well-defined orientation, and the high crystallinity and good orientation help to improve the electrical properties of the material. The IR patterns of BTONWs doped PVDF with different mass ratios as verified in Fig 3(g). The experimental analysis in this paper only took 1wt%, 2wt%, and 3wt% of BTO content as samples for comparison, because the more doping, the more agglomeration, which ultimately led to performance reduction [12]. Significant absorption peaks appear at about 1275 cm$^{-1}$ and 840 cm$^{-1}$, which correspond to the characteristic vibrations of the $\alpha$-phase and $\beta$-phase crystalline forms of PVDF, respectively. The intensity of the $\beta$-phase vibration peaks (1275 cm$^{-1}$) increased with the increase of the mass ratio of BTONWs, while the $\alpha$-phase vibration peaks (840 cm$^{-1}$) did not change much. The intensity of the $\beta$-phase vibration peaks increased while the $\alpha$-phase vibration peaks gradually weakened at 840 cm$^{-1}$. The $\beta$-phase content of PVDF increases and the $\alpha$-phase content decreases as the doping ratio of BTONWs increases. These IR spectral patterns indicated that the doping of BTONWs effectively promotes the crystallization of the $\beta$-phase of PVDF, thus enhancing the friction electrical properties of the material. The $\beta$-phase crystallization of PVDF was most significant at 3 wt% doping of BTONWs, indicating that the electrical properties of the material are optimal at this doping ratio. Through these IR spectra, The XRD patterns of Crystal characterization of BTONWs doped into PVDF with different mass ratios as shown in Fig 3(h). These XRD patterns indicated that the doping of BTONWs effectively promotes the crystallization of the $\beta$-phase of PVDF, which enhances the friction electrical properties of the material. The XRD showed that the intensity of BTONWs was detected in all phases and the successful doping of BTONWs into PVDF.

The performance metrics of a TENG with varying weight percentages of a material, likely BTONWs mixed with PVDF as revealed in Fig 4(a-c). The graphs illustrated the short-circuit current, open-circuit voltage, and transferred charge quantity over time for TENGs containing 0 wt%, 1 wt%, 2 wt%, and 3 wt% of the material. The TENG with 2 wt% achieved the highest short-circuit current, peaking around 12 μA, demonstrating the most significant enhancement in electrical output. The 2 wt% TENG showed the highest open-circuit voltage, peaking around 280V, indicating the best performance in this series. The 2 wt% TENG exhibited the highest charge transfer quantity, peaking near 110nC. The 2 wt% of BaTiO$_3$ in PVDF consistently provided the highest short-circuit current, open-circuit voltage, and transferred charge quantity, making it the optimal concentration for maximizing the TENG's electrical output. The 3 wt% concentration, while still improving over the baseline, didn't perform as well as the 2 wt%, suggesting that an excess of the material might hinder the performance improvements. This analysis underscored the importance of optimizing material concentrations in the design and fabrication of high-performance TENGs. When the nylon-11 and BaTiO$_3$/PVDF layers come into contact due to pressing, triboelectric electrification causes surface charge transfer. Nylon-11 loses electrons, becoming positively charged, while BaTiO$_3$/PVDF gains electrons, becoming negatively charged. Upon separation, a potential difference was created, driving electrons from the bottom to the top electrode, forming a negative current pulse. When pressed, a reverse positive current pulse was generated. This cycle produces an AC in the external circuit [13]. However, the output voltage was not AC and varies between 0 and $V_{OC}$, as explained by a capacitance model, resulting in only negative voltage amplitude due to the voltage varying from 0 to $V_{OC}$ [14,15].

The variation of the dielectric constant of BTONWs doped PVDF with different mass percentages as a function of frequency, ranging from 102 to 106 Hz as shown in Fig 4(d). The dielectric constants of PVDF were significantly increased with the increase in the doping ratio of BTONWs. This indicated that the doping of BTONWs can effectively enhance the electrical properties of PVDF. The trend of decreasing dielectric constant with increasing frequency at all doping ratios could be related to the agglomerate of a cluster. The aggregation reduced the effective surface area of the nanofibres, weakened

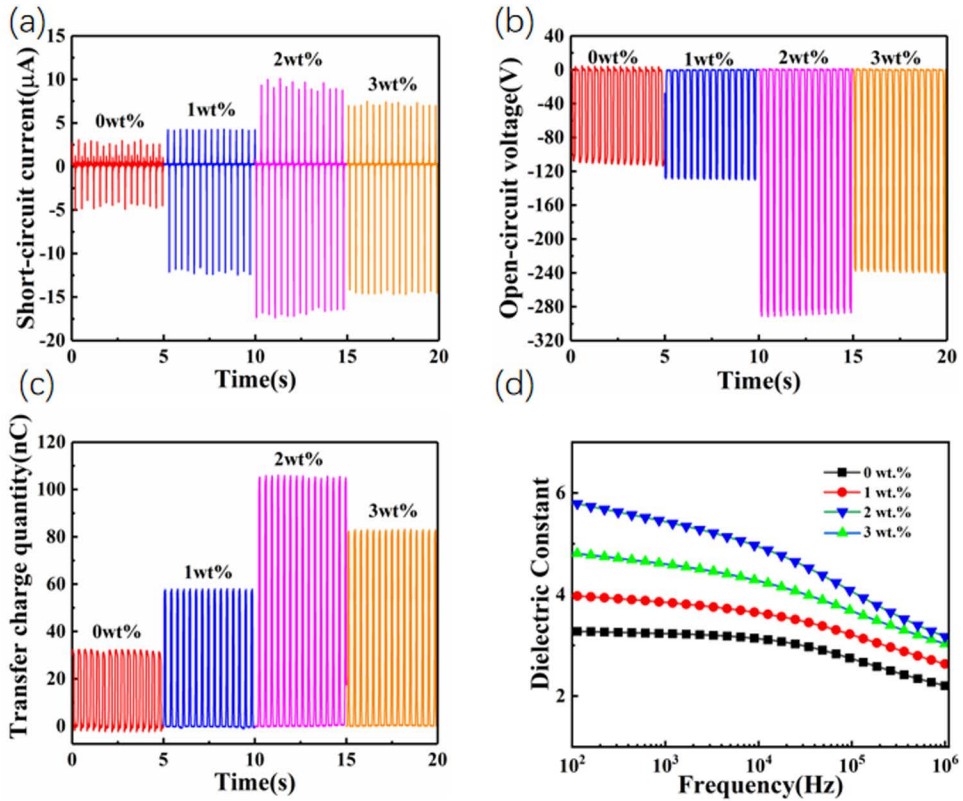

**Fig 4. BTONWs-PVDF electrical output image.** Image of **(a)** Short-circuit current. **(b)** Open-circuit voltage. **(c)** Transfer charge quantity. **(d)**Dielectric constant.

the interfacial polarisation effect, increased the conductivity loss, and led to inhomogeneity in the internal structure and electric field distribution of the material. At a BTONWs content of 2 wt%, the voltage output was higher, even though the dielectric constant was lower. This could be attributed to the fact that the charge separation and recombination processes were more optimized at this doping ratio, resulting in enhanced voltage generation. However, the higher dielectric constant did not lead to improved voltage output when the BTONWs content was increased to 3 wt%. Instead, the excessive presence of fillers causes increased contact or agglomeration, which hinders effective charge separation and transport, ultimately reducing the voltage. Although composites with 3 wt% BTONWs have a higher dielectric constant, this didn't necessarily translate to higher voltage output. The 2 wt% content might have achieved a more optimal balance of interfacial effects, charge separation efficiency, and mechanical properties, leading to better voltage performance [9]. The enhancement of material properties didn't depend only on a single physical quantity but was the result of a combination of factors.

Fig 5(a) shows the relationship between load resistance and both voltage and current. As the load resistance increases, the voltage initially rises, reaching a peak around 175V at an optimal resistance. Conversely, the current decreased with increasing resistance, indicating a trade-off between voltage and current. Fig 5(b) illustrates the power output as a function of resistance. The power output showed a sharp increase, peaking at around 1.45W at 90MΩ, and then gradually decreased as the resistance increased further. This confirmed the optimal resistance for maximum power output, highlighting the importance of matching the load resistance to the TENG for efficient energy harvesting. Fig 5(c) shows the voltage as a function of time under different patting speeds. Initially, the TENG was subjected to slow patting, resulting in lower and more spaced-out voltage peaks. When the patting speed increases, the voltage peaks become

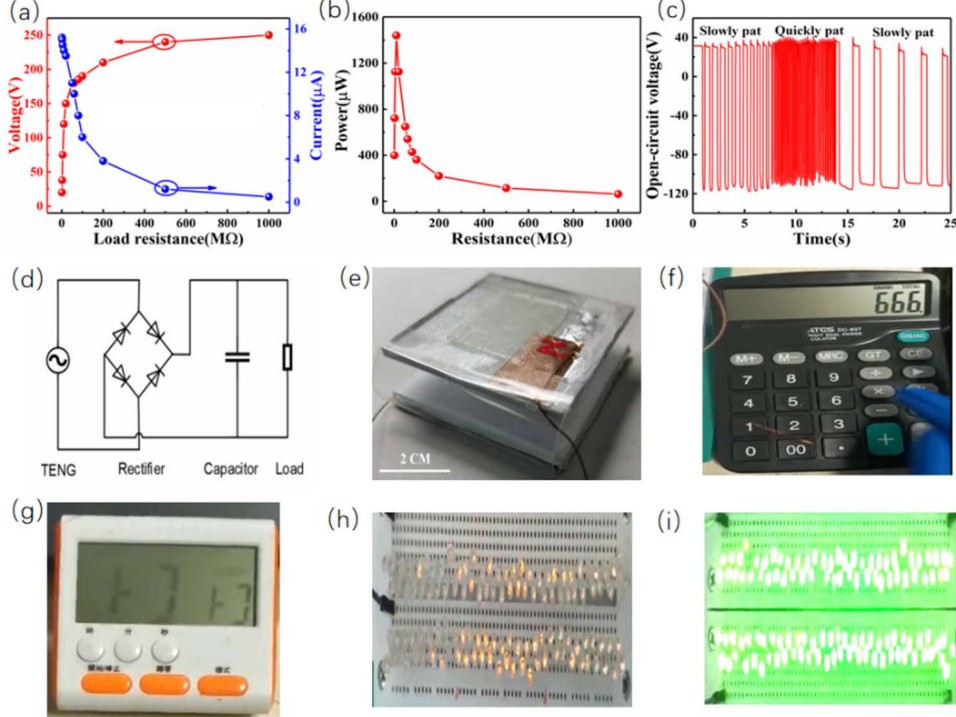

**Fig 5. Application of BTONWs-PVDF TENG as a power supply. (a)** Load output voltages (left axis), currents (right axis), and **(b)** instantaneous power densities of the optimized BTONWs-PVDF at load resistances of 0-1000 MΩ. **(c)** Different tapping times of the optimal TENG of BTONWs-PVDF related to measured voltage**(d)** External circuit diagram. **(e)** Physical diagram of TENG of BTONWs-PVDF. Photographs of **(f)** a calculator and **(g)** a timer in working condition. **(h)** A yellow LED and **(i)** a green LED to be lighted.

more frequent and higher in magnitude, demonstrating the TENG's responsiveness to varying mechanical inputs. Returning to slow patting showed a decrease in both the frequency and magnitude of the voltage peaks. The external circuit diagram used for the experiment is shown in Fig 5 (d), and Fig 5 (e) shows a physical diagram of TENG of BTONWs-PVDF. In this experiment, a TENG was tested for its capability to generate and store electrical energy by continuously patting the device. The TENG's performance was evaluated by its ability to power a calculator, a timer, and an array of LED lights. The objective was to demonstrate the practical application of TENG in powering everyday electronic devices. The initial step involved connecting the TENG to a calculator. The calculator, as shown in Fig 5(f), was initially off. After a series of pats on the TENG, the calculator powered on, displaying "666" on its screen. This indicated that the TENG successfully generated enough power to operate the calculator and was connected to a digital timer, as depicted in Fig 5(g). Similar to the calculator, the timer was initially off. After patting the TENG, the timer powered on, displaying the time "1:37." This further validated the TENG's capability to generate and store sufficient energy for practical use. Finally, TENG drives an array of LEDs by rectifying them. The yellow and green LED array before and after being powered by the TENG as shown in Fig 5 (h-i). Initially, the LEDs were off. Upon patting the TENG, the yellow and green LEDs lit up, demonstrating the TENG's ability to provide continuous and reliable power to light multiple LEDs. This part of the experiment showcased the TENG's potential for lighting applications, especially in low-power scenarios.

Through these experiments, the TENG demonstrated its efficiency in generating and storing energy, enough to power common electronic devices like calculators, timers, and LED lights. The results underlined the practical potential of TENG technology for sustainable and portable energy solutions in real-world applications. We also completed a comparison with traditional methods as shown in Table 1.

**Table 1. Comparison of different doping materials in TE layers.**

| TE layer | Size(cm²) | Voc(V) | Isc(µA) | Pd(W/m²) | TENG |
|---|---|---|---|---|---|
| PVDF-none | 2.0*2.0 | 22 | 0.6 | – | [16] |
| PVDF-ZnO | 2.5*3.0 | 119 | 1.6 | 0.11 | [17] |
| PVDF-MXene | 2.0*2.0 | 46 | 2.4 | 0.29 | [18] |
| PVDF-CoFe$_2$O$_4$ | 2.0*2.0 | 17.2 | 2.27 | 0.09 | [19] |
| PVDF-BaTiO$_3$(this work) | 4.0*4.0 | 280 | 12 | 1.45 | – |

## Conclusion

In summary, we developed a new method to prepare nanocomposite capacitors with high-performance energy harvesting and continuous output. Experimentally, it was demonstrated that BTONWs can improve the electrical output of nanocomposites. The dielectric constant increases by 54.2% at 2%wt doping compared to pure PVDF. To the best of our knowledge, this electrical output is the highest reported in the literature for previously PVDF-based TENGs. In addition, the nanocomposite material illuminates 120 LEDs, enables the calculator and timer to function through a simple external circuit, and illuminates the LEDs simultaneously when the TENG is continuously tapped. The experiments show that the new composite TENG can be used as a micro-generator and the fabric-based generator has good electrical properties. In conclusion, the nylon-11/BTONWs-PVDF micro-nanogenerator has the advantages of being lightweight, portable, self-powered, and inexpensive, which can have a good prospect for application in flexible sensors and wearable devices.

## Acknowledgments

The authors would like to thank the government of Chongqing, China, and SEGi University Malaysia for support.

## Author contributions

**Conceptualization:** Xiong Dien, Yang Zhuanqing.

**Data curation:** Xiong Dien, Cheng Linfeng, Yuan Jiang.

**Formal analysis:** Xiong Dien, Cheng Linfeng.

**Funding acquisition:** Yang Zhuanqing.

**Methodology:** Xiong Dien, Zhang Lin.

**Project administration:** Nurulazlina Ramli, Tzer Hwai Gilbert Thio.

**Resources:** Yang Zhuanqing.

**Software:** Zhang Lin.

**Supervision:** Nurulazlina Ramli, Tzer Hwai Gilbert Thio, Zhang Lin.

**Visualization:** Yuan Jiang.

**Writing – original draft:** Xiong Dien, Cheng Linfeng.

**Writing – review & editing:** Nurulazlina Ramli, Tzer Hwai Gilbert Thio.

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
