## [Decision Letter · Decision Letter 0]

2 Dec 2024

PONE-D-24-50697High-Performance Energy Harvesting and Continuous Output Using nylon-11/BaTiO3-PVDF Triboelectric Nanogenerators with Strong Dielectric PropertiesPLOS ONE

Dear Dr. Dien,

Thank you for submitting your manuscript to PLOS ONE. After careful consideration, we feel that it has merit but does not fully meet PLOS ONE’s publication criteria as it currently stands. Therefore, we invite you to submit a revised version of the manuscript that addresses the points raised during the review process.

**Please see the attached comments of all the reviewers, ** the submitted manuscript doesn't include sufficient experimental results to validate the performance of the proposed work. Therefore, the authors are suggested to provide more experimental data.==============================

We look forward to receiving your revised manuscript.

Kind regards,

Azim Uddin, Ph.D.

Academic Editor

PLOS ONE

Journal requirements: When submitting your revision, we need you to address these additional requirements. 1. Please ensure that your manuscript meets PLOS ONE's style requirements, including those for file naming. The PLOS ONE style templates can be found at https://journals.plos.org/plosone/s/file?id=wjVg/PLOSOne_formatting_sample_main_body.pdf and https://journals.plos.org/plosone/s/file?id=ba62/PLOSOne_formatting_sample_title_authors_affiliations.pdf 2. Please amend either the title on the online submission form (via Edit Submission) or the title in the manuscript so that they are identical. 3. Please include a caption for figure 3. 4. PLOS requires an ORCID iD for the corresponding author in Editorial Manager on papers submitted after December 6th, 2016. Please ensure that you have an ORCID iD and that it is validated in Editorial Manager. To do this, go to ‘Update my Information’ (in the upper left-hand corner of the main menu), and click on the Fetch/Validate link next to the ORCID field. This will take you to the ORCID site and allow you to create a new iD or authenticate a pre-existing iD in Editorial Manager. 5. We note that the grant information you provided in the ‘Funding Information’ and ‘Financial Disclosure’ sections do not match.  When you resubmit, please ensure that you provide the correct grant numbers for the awards you received for your study in the ‘Funding Information’ section. 6. Thank you for stating the following in the Acknowledgments Section of your manuscript: [This work was supported by Natural Science Foundation Project of Chongqing Science & Technology Commission through grant No.2024NSCQ-MSX4013, and Scientific and Technological Research Program of Chongqing Municipal Education Commission through grant No.KJQN202403414.]We note that you have provided funding information that is not currently declared in your Funding Statement. However, funding information should not appear in the Acknowledgments section or other areas of your manuscript. We will only publish funding information present in the Funding Statement section of the online submission form. Please remove any funding-related text from the manuscript and let us know how you would like to update your Funding Statement. Currently, your Funding Statement reads as follows:  [The author(s) received no specific funding for this work.] Please include your amended statements within your cover letter; we will change the online submission form on your behalf.

Reviewers' comments:

Reviewer's Responses to Questions

**Comments to the Author**

1. Is the manuscript technically sound, and do the data support the conclusions?

Reviewer #1: No

Reviewer #2: Yes

Reviewer #3: Yes

2. Has the statistical analysis been performed appropriately and rigorously? 

Reviewer #1: No

Reviewer #2: Yes

Reviewer #3: Yes

3. Have the authors made all data underlying the findings in their manuscript fully available?

Reviewer #1: Yes

Reviewer #2: Yes

Reviewer #3: Yes

4. Is the manuscript presented in an intelligible fashion and written in standard English?

Reviewer #1: Yes

Reviewer #2: Yes

Reviewer #3: Yes

5. Review Comments to the Author

Reviewer #1: This manuscript does not include sufficient experimental results to validate the performance of the proposed TENG. This version of the manuscript must be significantly improved based on the following issues:

1.- The introduction should include the advantages and limitations of the TENGs reported in the literature. Furthermore, this section must consider the scientific contribution, advantages, or novelty of the proposed TENG.

2.- This manuscript must consider the design phase of the proposed TENG, considering detailed descriptions (e.g., figures) of its dimensions, materials, and performance.

3.-The authors must regard more detailed information on the fabrication process of the TENG.

4.- The size of Figure 3(f,g,h) is small. In addition, the resolution of this Figure must be enhanced.

5.- The resolution and quality of Figures 4 and 5 must be improved.

6.- This manuscript requires more experimental tests of the stability and degradation of the performance of the TENG.

7.- This manuscript needs experimental results on the effect of the relative humidity on the output voltage of the TENG.

8.- The authors should include a table with the main parameters, advantages, and limitations of the proposed TENG compared to other types of recent TENGs reported in the literature.

9.- What are the main challenges of the proposed TENG?

10.- What are the future research work?

11.- The conclusions must be enhanced based on the previous comments.

Reviewer #2: nylon-11/BaTiO₃-PVDF Triboelectric Nanogenerators were showcased in this work. It achieved a current and voltage of 12 μA and 280 V. After seeing this work I gave a minor revision after following these comments such as:

1. Why dielectric constant at higher wt% doping decreases but in general it increases. Explain properly?

2. In XRD after 20 there is additional peak which is not index must be given reason for occurrence. The lattice parameters must be given properly? Charging of capacitor and discharge for Figure 5 g, h, i can be added?

3. Working mechanism has some error the arrow direction for flow of current is same., Also added the arrow for flow of electrons as well.

4. Electrical output of TENG must be optimize at various air gap. How mechanical properties of the composite effect the performance of TENG.

5. Why PVDF was chosen inspite of PVDF TrFE as beta phase is proven to be higher in PVDF-TrFE. How the authors optimize this electric field for there electrospinning 21 KV?

6. Important references in TENG need to be added such as: Nano Energy 131, 110319, 2024; Journal of Materiomics 10 (4), 792-802, 2024; Journal of Science: Advanced Materials and Devices 9 (2), 100693, 2024 and Nano Energy 101, 107620, 2022.

7. English grammertical errors need to be removed.

Reviewer #3: The article titled "High-Performance Energy Harvesting and Continuous Output Using nylon-11/BaTiO₃-PVDF Triboelectric Nanogenerators with Strong Dielectric Properties" presents a significant contribution to the field of triboelectric nanogenerators (TENGs) by optimizing BaTiO₃ doping in PVDF for improved energy conversion. The results are interesting; however, the article requires some improvements and corrections. I suggest that the manuscript can be considered for publication in PLOS ONE Journal after minor revisions.

The suggested improvements and corrections are:

Abstract section

1. Could the authors provide quantitative comparisons with existing TENG systems to highlight the novelty of achieving a power density of 1.45 W/m²?

2. Why was 2 wt% BaTiO₃ selected as the optimal doping ratio, and how does it compare with other ratios in terms of performance trade-offs?

Introduction section

3. Could the authors elaborate on the challenges associated with scaling TENG technology for commercial applications, especially in terms of material costs and production feasibility?

4. How does the authors findings address existing gaps in TENG performance and durability for wearable and low-power electronic applications?

Materials and Methods section

5. Could the authors include the environmental conditions (e.g., temperature, humidity) under which TENG performance was tested to ensure reproducibility?

6. Could the authors explain the rationale for excluding BaTiO₃ doping ratios above 3 wt%, and were any preliminary tests conducted at higher ratios?

7. Can the authors provide more details about the electrospinning parameters, such as the material of the syringe and adjustments made to flow rates and drum speed during the fabrication process?

8. How does the environmental sustainability of your materials and processes compare to other TENG fabrication methods?

Results and Discussion section

9. Could the authors improve the clarity and resolution of Figures 3 and 5 to make the structural details and annotations more legible?

10. Would the authors consider adding a comparative table or schematic summarizing the electrical performance of TENGs with different material compositions for easier reference?

11. Could the authors expand on the interaction between the dielectric constant, mechanical stability, and electrical output at different BaTiO₃ doping ratios, especially above 2 wt%?

12. Have the authors tested the long-term stability and degradation of the TENG under continuous operation, and if so, could you discuss the results?

13. Can the authors provide further insights into the scalability of your approach for large-scale production, including potential challenges or cost analyses?

14. How do the authors envision the practical implementation of your TENG for powering wearable devices or low-power electronics, given the current state of the technology?

Language and Formatting

15. Would the authors consider correcting typographical errors (e.g., "couldnot" to "could not" and “probles” in the 4th paragraph in the introduction section) and ensuring consistent use of technical terms throughout the manuscript?

16. Could the authors ensure the figure captions align with the journal’s formatting guidelines for clarity and consistency?

17. Could the authors clarify the abbreviation “PDMS and PMMA” in the beginning of the manuscript.

References section

18. Are all your references up-to-date, particularly regarding recent advancements in triboelectric nanogenerators and their applications?

19. Could you include more references on the scalability and commercialization of TENGs to provide additional context for your study?

6. PLOS authors have the option to publish the peer review history of their article (what does this mean? ). If published, this will include your full peer review and any attached files.

**Do you want your identity to be public for this peer review?** For information about this choice, including consent withdrawal, please see our Privacy Policy .

Reviewer #1: No

Reviewer #2: No

Reviewer #3: No

---

## [Author Response · Author response to Decision Letter 0]

23 Jan 2025

Response to Reviewer 1

Comment-1

The introduction should include the advantages and limitations of the TENGs reported in the literature. Furthermore, this section must consider the scientific contribution, advantages, or novelty of the proposed TENG.

Response: We feel great thanks for your professional review work on our article. According to your nice suggestions, We have corrections to our previous draft in introduction with red:

TENGs limit their use due to lower output power density, and low conversion efficiency makes it difficult to drive loads(2024,Micromachines, 15(9), 1114).

BaTiO₃promotes better crystallization of PVDF. in addition to its own good electrical properties, BaTiO₃ has better performance as a composite(2024,Nano Energy, 122, 109264).

Specifically, this paper presented the fabrication a new TENG of high-performance, highly flexible using nylon-11 and PVDF with BaTiO₃NWs.

Comment-2

This manuscript must consider the design phase of the proposed TENG, considering detailed descriptions (e.g., figures) of its dimensions, materials, and performance.

Response: We sincerely thank the editor and all reviewers for their valuable feedback that we have used to improve the quality of our manuscript. We have corrected it in the manuscript with red. Fig. R1(a) shows a schematic diagram of the device and the fabricated TENG. When the nylon-11 and the PVDF-BTONWs are contacted under pressure, a surface charge transfer occurs in the contact area due to the triboelectric effect, where the positively charged layer of the nylon-11 (top) loses electrons to carry a positive charge, while the negatively charged layer of the PVDF-BTONWs (bottom) gains electrons to carry a negative charge. After the separation of the nylon-11 and the PVDF-BTONWs, a potential difference between the top and bottom electrodes occurs due to electrostatic induction, which induces electrons to flow from the bottom electrode to the top electrode, forming a half-cycle negative current pulse. When the nylon-11 and PVDF-BTONWs are pressed, a reverse current pulse, i.e., a half-cycle positive current pulse, is generated. Repeatedly, the external circuit could generate AC during each pressing and releasing cycle, as shown in Fig. R1(b). The most TENGs structure are the vertical separation type(Nano Energy, 2024,122, 109264)(Results in Materials,2024, 22, 100576).

Fig. R1. (a) Schematic diagram of the device assembly structure. (b) The working mechanism of TENG.

Than, The BTONWs-PVDF nanofibers were deposited on an aluminum foil using electrostatic spinning, and the resulting sample was cut into 4 × 4 cm² squares. Subsequently, the squares were fixed to laser-cut 3 mm thick acrylic substrates by using double-sided adhesive tape to tightly fit the side with the nanofibers (the aluminum foil as the backing layer) to the acrylic plate, so that the aluminum foil served as both a mechanical support layer and an electrode. In order to induce an electrical signal, a copper wire with a diameter of about 0.51 mm was connected to the aluminum foil, and a transparent PET film with a thickness of 0.5 mm was used as a contact-separation friction interface (as shown in Fig.R2.). This designed structure not only simplifies the device assembly steps, but also lays the foundation for subsequent performance evaluation and application testing.

Fig. R2. (a)-(d) The preparation process of BTONWs-PVDF nanofibers. (e) The designed of TENG. (f) Digital picture of assembled TENG.

PVDF exhibits exceptional polarization capabilities, which contribute to enhancing the storage and transfer efficiency of triboelectric charges. It possesses a high dielectric constant, excellent flexibility, and mechanical properties, enabling it to adapt to various deformation requirements, making it well-suited for wearable devices and flexible electronics. Furthermore, PVDF demonstrates strong resistance to humidity, temperature fluctuations, and chemical corrosion, ensuring its suitability for long-term applications. Due to its strong negative triboelectric properties, PVDF has been widely used in TENG devices to effectively capture positively charged triboelectric charges, thereby improving output performance (Advanced Materials, 2019, 31(40), 1808487).

Nylon-11 exhibits favorable triboelectric properties, displaying significant positive triboelectric polarity when in contact with negatively charged materials. It also possesses excellent wear resistance and toughness, enabling it to maintain stable performance under repeated friction. Nylon-11 is widely employed as the positive electrode material in TENGs due to its pronounced positive polarity, forming a significant potential difference with PVDF, which enhances the efficiency of triboelectric charge separation (Nano Energy, 2017, 37, 1–24).

The electrical performance was measured using a Keithley 6514 electrometer, which is highly sensitive and particularly suitable for detecting weak signals (Advanced Energy Materials, 2017, 7(23)). The output power was calculated in real time based on the formula P=UI, where the measured voltage (U) and current (I) were used for the calculation(Advanced Materials 2023, 35, 2208139; Advanced Materials Technology 2019, 1900905; Microelectronic Engineering 2022, 257, 111725).

Comment-3

The authors must regard more detailed information on the fabrication process of the TENG.

Response: We think this is an excellent suggestion. Below is the detailed fabrication process of the TENG, which we have partially modified in the original draft with red.In this study, BTONWs-PVDF nanofibers were first deposited on aluminum foil using electrostatic spinning. Pour the precursor into a 10 mL syringe made with a polypropylene barrel, 21 G stainless steel needle, and PTFE tubing. Injections were made at a rate of 1 ml per hour on a syringe pump. The needle was mounted on a reciprocating moving platform and a voltage of 21 KV was applied to the needle with a distance of 17 cm from the needle to the drum collector for collection, and the drum collector was rotated at 1500 rpm. The resulting samples were cut into 4x4 cm². Subsequently, the square was fixed on a laser-cut 3 mm thick acrylic resin substrate, and the side with the nanofibers (the aluminum foil as the bottom layer) was tightly attached to the acrylic plate using double-sided adhesive, so that the aluminum foil served as both a mechanical support layer and an electrode. In order to induce an electrical signal, a copper wire with a diameter of about 0.51 mm was connected to the aluminum foil, and a transparent PET film with a thickness of 0.5 mm was used as a contact-separation friction interface (as shown in Fig. R3.). Finally, the copper wire was connected to a Keithley Model 6514 electrical test system to characterize the electrical output of the device, including parameters such as current and voltage. This designed structure not only simplifies the device assembly steps, but also lays the foundation for subsequent performance evaluation and application testing.

Fig. R3.Physical image of TENG.

Comment-4

The size of Figure 3(f,g,h) is small. In addition, the resolution of this Figure must be enhanced.

Response: According to reviewers comments, Considering that Fig. 3 consists of multiple figures, we performed a reformatting and made corrections in the manuscript as shown in Fig. R4. Thank you again for your positive comments and valuable suggestions to improve the quality of our manuscript.

Fig. R4.Analytical image of TENG produced by nylon 11 and PVDF-BTONWs. (a) Physical image of TENG. (b) Surface morphology of nylon-11 fibres .(c) Surface morphology of BTONWs -PVDF. (d)SEM images of BTONWs doped with different mass ratios. (e)Magnified images of BTONWs doped with different mass ratios. (f)XRD image of BTONWs. (g) XRD patterns of PVDF, BTONWs-PVDF film, and BTONWs before mixing with PVDF. (h)FTIR absorption spectra of BTONWs-PVDF nanofibres with different BTONWs mass fractions.

Comment-5

The resolution and quality of Figures 4 and 5 must be improved.

Response: We sincerely thank the reviewer for careful reading. We entered corrected Figures 4 and 5 in the manuscript.

Comment-6

This manuscript requires more experimental tests of the stability and degradation of the performance of the TENG.

Response: We sincerely thank the reviewer for comments. We agree with the reviewer that practical environmental factors will affect the performance of TENGs. In this work, similar to what has been usually done in the literature, we have measured the voltage stability of nylon11/PVDF-BTONWs based TENG under lab environment, as shown in Fig. R5, which has constant temperature and humidity, for ~30000 cycles, showing its stability.

Fig. R5. The stability of nylon11/PVDF-BTONWs based TENG

Comment-7

This manuscript needs experimental results on the effect of the relative humidity on the output voltage of the TENG.

Response:We feel great thanks for your professional review work on our article. According to your nice suggestions,We have done experiments related to the effect of humidity and temperature on TENG. Fig. R6. demonstrates the effect of humidity and temperature on the output performance of TENG. (a) illustrates the variation of TENG output voltage over time under humidity levels of 30%, 50%, 70%, and 90%. It can be observed that as humidity increases, the output voltage gradually decreases. This is attributed to the higher humidity levels causing water molecules in the air to form a conductive water film on the material surface, leading to rapid leakage of surface triboelectric charges and weakening the triboelectric effect (Small, 2024, 2401846). (b) shows the output current under different humidity conditions. Similarly, the current significantly decreases in high-humidity environments, further confirming that increased humidity causes charge leakage and reduces charge transport efficiency (Small, 2024, 2401846). (c) depicts the trend of charge quantity with humidity. It can be observed that higher humidity levels result in a notable reduction in charge accumulation, indicating that high humidity suppresses the charge storage capacity of the TENG, thereby reducing its overall output performance (Small, 2024, 2401846). (d)–(f) illustrate the changes in TENG output voltage, current, and charge quantity at temperatures of 30°C, 60°C, and 90°C. As the temperature increases, the TENG maintains stable output performance within a certain range. Particularly at higher temperatures, the TENG output waveforms exhibit good periodicity and stability. The elevated temperature may enhance the flexibility and surface activity of the materials, thereby improving the contact quality between the materials and partially compensating for the electrical performance losses caused by temperature changes (Advanced Engineering Materials, 2017, 19(12), 1700275). This indicates that TENGs have a certain degree of stability and adaptability under fluctuating temperature conditions.

Fig. R6. Effect of Humidity and Temperature on the Electrical Output of TENGs. (a)-(c)Effect of humidity at 30%, 50%, 70%, and 90% on the voltage, current, and charge of TENGs. (d)-(f)Effect of temperature at 30, 60, and 90 on voltage, current, and charge of TENGs.

Comment-8

The authors should include a table with the main parameters, advantages, and limitations of the proposed TENG compared to other types of recent TENGs reported in the literature.

Response: We sincerely thank the reviewer for your comments. As shown in Table 1, we have updated the table, which we can see that PVDF-BaTiO3 has better and superior output performance compared to the previously reported PVDF-based TENGs and modified in the manuscript.

Table 1 Comparison of different dopant materials in TE layers.

TE layer Size(cm2) Voc(V) Isc(μA) Pd�W/m2� TENG

PVDF-none 2.0*2.0 22 0.6 - [29]

PVDF-ZnO 2.5*3.0 119 1.6 0.11 [30]

PVDF-MXene 2.0*2.0 46 2.4 0.29 [31]

PVDF-CoFe2O4 2.0*2.0 17.2 2.27 0.09 [32]

PVDF-BaTiO3(this work) 4.0*4.0 280 12 1.45 -

Comment-9

What are the main challenges of the proposed TENG?

Response: We feel great thanks for your professional review work on our article. Overall, there is a great market potential to commercialise TENG, which can be significantly reduced through low-cost material development, process optimisation, environmental adaptation improvements and market positioning adjustments. From a technical point of view, TENG already has a relatively established technology, and the spraying technique has been shown to be more cost-effective in batch production while maintaining similar nanostructured properties (Advanced Materials Interfaces, 2019, 6(15), 1900557). Similarly, the addition of graphene-reinforced composites not only improves wear resistance, but also extends the lifetime to 1 million cycles of operation (Nano Energy, 2020, 67, 104258). In terms of materials, expensive polymers can be replaced with natural materials such as cellulose, chitosan and other bio-based materials. Recent studies have shown that TENGs prepared by using discarded plastic bottles or milk cartons not only have superior performance, but also have extremely low raw material costs (Environmental Science & Technology, 2021, 55(6), 3712-3721). The challenge is mainly in the lack of stability of TENG. As researchers continue to study, TENG is expected to achieve large-scale applications in wearable devices, smart homes and industrial monitoring.

Comment-10

What are the future research work?

Response:We feel great thanks for your professional review work on our article again. Although the basic material of TENG is completed in the text, using Nylon 11 with PVDF-BTONWs, it is possible to make improvements in the structural aspects and stability of TENG based on recent literature. For example, making a porous structure improves the effective contact surface, and through chemical modification, researchers can introduce functional groups with higher friction electrical sequence positions, which improves the material's charge generation capability. And through surface roughening, the contact area of the material can be increased to further improve the charge transfer efficiency.In 2021, Van-Tien Bui et al. fabricated a bird's nest structure of TENG, and the altered structure doubled the output electrical properties (Nano Energy,2020, 71, 104561). In the same year, Cheedarala used sand to polish the surface of the friction layer, which improved its electrical properties by 40%, but this structure was inferior to the hole structure due to the modification of the surface structure only (International Journal of Smart and Nano Materials, 2019, 11(1), 9). Doldet Tantraviwat listed about the effect of the three types of structures, namely, holes, columns, and flakes, on the PDMS-based TENG, and the experimental results showed that the hole-containing structure has the most optimised electrical output of TENG (Nano Energy ,2019, 67). Secondly, the durability can be increased by spraying compounds, etc. TENGs are prone to failure of the friction electric layer during constant contact separation, which is due to the fact that the friction layer is easy to stick together with the fibres formed by electrostatic spinning, so the TENGs usually decay within a short period of time during the pressing process. Xiang Gang et al. proposed a TENG made of ethyl cellulose wrapped nylon11 and PTFE wrapped PVDF, which can resist 100,000 presses and have significantly enhanced mechanical strength (ACS Applied Materials And Interfaces, 2023 15.45:52696-52704). In addition, Yoonsang Ra was sandblasted with a metallic layer on the friction surface, with an amazing durability of 150,000 presses (Environ. Technol. Innov. 2021, 24, 102049).

Comment-11

The conclusions must be enhanced based on the previous comments

Response: We thank the reviewer for your comments. We tried our best to improve the conclusions. And we marked in red in the revised paper. In summary, we developed a new method to prepare nanocomposite capacitors with high performance energy harvesting and continuous output. Experimentally, it was demonstrated that BTONWs can improve the electrical output of nanocomposite

---

## [Decision Letter · Decision Letter 1]

10 Feb 2025

High-performance energy harvesting and continuous output using nylon-11/BaTiO₃-PVDF triboelectric nanogenerators with strong dielectric properties

PONE-D-24-50697R1

Dear Dr. Zhuanqing,

We’re pleased to inform you that your manuscript has been judged scientifically suitable for publication and will be formally accepted for publication once it meets all outstanding technical requirements.

Kind regards,

Azim Uddin, Ph.D.

Academic Editor

PLOS ONE

Additional Editor Comments (optional):

Reviewers' comments:

Reviewer's Responses to Questions

**Comments to the Author**

1. If the authors have adequately addressed your comments raised in a previous round of review and you feel that this manuscript is now acceptable for publication, you may indicate that here to bypass the “Comments to the Author” section, enter your conflict of interest statement in the “Confidential to Editor” section, and submit your "Accept" recommendation.

Reviewer #2: All comments have been addressed

Reviewer #3: All comments have been addressed

2. Is the manuscript technically sound, and do the data support the conclusions?

Reviewer #2: Yes

Reviewer #3: Yes

3. Has the statistical analysis been performed appropriately and rigorously? 

Reviewer #2: Yes

Reviewer #3: Yes

4. Have the authors made all data underlying the findings in their manuscript fully available?

Reviewer #2: Yes

Reviewer #3: Yes

5. Is the manuscript presented in an intelligible fashion and written in standard English?

Reviewer #2: Yes

Reviewer #3: Yes

6. Review Comments to the Author

Reviewer #2: The authors work has been improved after revision. The authors have added new references as well to support the study. Hence I accept it as it is.

Reviewer #3: I have reviewed the revised version of the manuscript titled "High-performance energy harvesting and continuous output using nylon-11/BaTiO₃-PVDF triboelectric nanogenerators with strong dielectric properties" and appreciate the authors' efforts in addressing the comments and improving the quality of their work. The revisions effectively respond to the concerns raised in the initial review, and the manuscript now meets the standards for publication in PLOS ONE.

Based on my assessment, I recommend the acceptance of this manuscript.

7. PLOS authors have the option to publish the peer review history of their article (what does this mean? ). If published, this will include your full peer review and any attached files.

**Do you want your identity to be public for this peer review?** For information about this choice, including consent withdrawal, please see our Privacy Policy .

Reviewer #2: No

Reviewer #3: **Yes: ** Mohamed M. Salem

---

## [Editor Report · Acceptance letter]

PONE-D-24-50697R1

PLOS ONE

Dear Dr. Zhuanqing,

I'm pleased to inform you that your manuscript has been deemed suitable for publication in PLOS ONE. Congratulations! Your manuscript is now being handed over to our production team.

Kind regards,

on behalf of

Dr. Azim Uddin

Academic Editor

PLOS ONE